# Promotion of neutralizing antibody-independent immunity to wild-type and SARS-CoV-2 variants of concern using an RBD-Nucleocapsid fusion protein

Julia T. Castro[1,2,3], Patrick Azevedo[1,2], Marcílio J. Fumagalli[3], Natalia S. Hojo-Souza[1,2], Natalia Salazar[1], Gregório G. Almeida[2], Livia I. Oliveira[2,4], Lídia Faustino[2], Lis R. Antonelli[2], Tomas G. Marçal[2], Marconi Augusto[4], Bruno Valiate[2], Alex Fiorini[1], Bruna Rattis[3], Simone G. Ramos[3], Mariela Piccin[3], Osvaldo Campos Nonato[3], Luciana Benevides[3], Rubens Magalhães[1], Bruno Cassaro[1], Gabriela Burle[1,2], Daniel Doro[1,2], Jorge Kalil[5], Edson Durigon[6], Andrés Salazar[7], Otávia Caballero[7], Helton Santiago[1,8], Alexandre Machado[1,2], João S. Silva[3], Flávio da Fonseca[1,8], Ana Paula Fernandes[1,9], Santuza R. Teixeira[1,8] & Ricardo T. Gazzinelli[1,2,3,10] ✉

Both T cells and B cells have been shown to be generated after infection with SARS-CoV-2 yet protocols or experimental models to study one or the other are less common. Here, we generate a chimeric protein (SpiN) that comprises the receptor binding domain (RBD) from Spike (S) and the nucleocapsid (N) antigens from SARS-CoV-2. Memory CD4[+] and CD8[+] T cells specific for SpiN could be detected in the blood of both individuals vaccinated with Coronavac SARS-CoV-2 vaccine and COVID-19 convalescent donors. In mice, SpiN elicited a strong IFN-γ response by T cells and high levels of antibodies to the inactivated virus, but not detectable neutralizing antibodies (nAbs). Importantly, immunization of Syrian hamsters and the human Angiotensin Convertase Enzyme-2-transgenic (K18-ACE-2) mice with Poly ICLC-adjuvanted SpiN promotes robust resistance to the wild type SARS-CoV-2, as indicated by viral load, lung inflammation, clinical outcome and reduction of lethality. The protection induced by SpiN was ablated by depletion of CD4[+] and CD8[+] T cells and not transferred by antibodies from vaccinated mice. Finally, vaccination with SpiN also protects the K18-ACE-2 mice against infection with Delta and Omicron SARS-CoV-2 isolates. Hence, vaccine formulations that elicit effector T cells specific for the N and RBD proteins may be used to improve COVID-19 vaccines and potentially circumvent the immune escape by variants of concern.

Since the end of 2019, over 500 million cases and six million deaths from COVID-19 have been reported worldwide. Most if not all COVID-19 vaccines, currently in use, are based on the Spike (S) protein and neutralizing antibodies (nAbs)[1,2]. However, positive selection of SARS-CoV-2 mutants with amino acid changes in the receptor binding domain (RBD) and adjacent segments from S protein is a frequent event[3,4] that generates the variants of concern (VOCs). The VOCs have conformational changes in the RBD, augmenting their affinity to the Angiotensin Convertase Enzyme-2 (ACE-2) and the ability to escape the action of nAbs[5]. While maintaining fitness, these conformational changes in the S protein are often associated with enhanced virus infectivity and spread in humans[6,7]. Indeed, the efficacy of the current vaccines targeting conformational RBD epitopes has been challenged by the emergence of VOCs[1,6,8–12].

The nAbs bind to the RBD from Spike (S) protein and prevent the interaction of SARS-CoV-2 with the ACE-2 and posterior host cell invasion[13]. While the levels of nAbs elicited by vaccination or infection are taken as the main predictors of protective immunity, a causal-effect relationship remains to be established[14–16]. Importantly, evidences for the function of T cells in mediating immunity to SARS-CoV-2 have accumulated[17]. For instance, asymptomatic patients have low, often undetectable, levels of anti-SARS-CoV-2 nAbs, whereas patients with moderate or severe COVID-19 display intermediate to high levels of circulating nAbs[18–25].

A coordinated response of CD4+ T cells, CD8+ T cells and nAbs seems to be ideal for a favorable outcome of disease[14,26]. Multiple T cell epitopes have been identified in SARS-CoV-2 proteins, some of which present homology to polypeptides from other coronavirus that circulate in the human populations and might explain the resistance of some seronegative individuals to symptomatic COVID-19[16,27–29]. Together, these studies suggest an important involvement of effector T cells in mediating resistance to primary infection with SARS-CoV-2[30]. However, most vaccines developed to protect against COVID-19 are based on conformational epitopes from the S protein and the induction of nAbs.

Here, we show that a chimeric protein containing the unfolded RBD from the canonical Spike protein and N protein (SpiN) is recognized by IgG antibodies and induces IFN-γ production by CD4+ and CD8+ T cells from convalescent and vaccinated human donors. In mice, we demonstrate that SpiN induces strong humoral and T cell responses but no detectable nAbs. Regardless, the SpiN-immunized mice become highly resistant to the wild type, the Delta, and the Omicron variants of SARS-CoV-2. Thus, we provide a model that can be used to explore the function of effector CD4+ and CD8+ T cells targeting conserved epitopes from SARS-CoV-2 and potentially circumvent the immune escape of VOCs.

## Results

### In silico analysis of T cell epitopes and mutations in the RBD and N proteins from SARS-CoV-2

First, we performed a silico analysis of the N and S proteins to identify the regions that are enriched for T cell epitopes[31,32]. As shown in Fig. 1a, the vertical lines below the bars representing the S and N proteins indicate each of the putative CD8+ T (red lines) and CD4+ T (black lines) cell epitopes for human, respectively. The results presented in the Supplementary Tables 1–4 shows the epitope sequences in RBD and N for HLA-I, HLA-DR and mice MHC-I and II. Consistent with a previous study, we found that the N protein is highly enriched for T cell epitopes[29]. The results presented in Supplementary Table 1 (Percentile Rank column) are the binding scores from IEDB and NETMHCII to validate the selected T cell epitopes. We included in the table only the peptides that presented scores < 1 for HLA-ABC and HLA-DR; < 1.3 for mouse MHC I; and < 2.0 for mouse MHC II. Based on this virtual analysis, the N protein has 32 immunogenic peptides with higher affinity for HLA-ABC (Supplementary Table 1), recognized by CD8+ T cells, and

11 for HLA-DR (Supplementary Table 2), recognized by CD4+ T cells. In the S protein, the RBD segment has the greatest prevalence of potential T cell epitopes, presenting 10 and 8 epitopes with high affinity to HLA-ABC (Supplementary Table 1) and HLA-DR (Supplementary Table 2), as determined by the binding score. Accordingly, both N and RBD proteins have been shown immunogenic for CD4+, CD8+ T cells and B lymphocytes[16,26,28,33–35].

Next, we analyzed the N and S protein sequences of the wild-type Wuhan isolate and five VOC lines (Alpha-B.1.1.7, Beta-B.1.351, Gama-P.1, Delta, and Omicron) distributed worldwide[36]. The peaks with circles indicate the position of most frequent amino acid changes in the N and S proteins, and the height of the peaks indicates the frequency that these changes occur in the Wuhan and the five VOC lines. The blue and purple circles indicate the most frequent amino acid changes observed in the VOCs. Importantly, only two T cell epitopes found in the N protein overlapped with sites of amino acid changes that are associated with the variants (peaks with purple circles) (Fig. 1a). These findings suggest that the majority of the putative T cell epitopes from the N protein are conserved in the VOCs. Furthermore, the N protein is highly expressed in the cytosol of host cells[37], and thus likely to be readily available for processing and presentation via HLA-I to the cytotoxic CD8 T cells. Hence, the N protein seems an ideal antigen for a T cell-based vaccine that has a broad effect against variants.

### Antibody and T cell responses in COVID-19 convalescent and vaccinated individuals

Aiming to develop a vaccine that induces a strong T cell-mediated immunity, we constructed a fusion protein, named SpiN, that bears the RBD from Spike and the Nucleocapsid (N) protein from SARS-CoV-2. Polyacrylamide gels (Supplementary Fig. 1a) and Western blots (Supplementary Fig. 1b, c) show the highly purified N (~45 kDa) and RBD (~25 kDa) proteins expressed in *Escherichia coli* as well as S surface antigen (~200 kDa) obtained from eukaryotic cells. The recombinant SpiN, also derived from bacteria, has an apparent molecular weight of 70 kDa (Supplementary Fig. 1a–c). The proteins were purified either on a nickel column (RBD and N) or by ion exchange chromatography (SpiN). The immunoblots were generated with either rabbit polyclonal anti-N (Supplementary Fig. 1b) or anti-RBD (Supplementary Fig. 1c) sera, and show that SpiN protein is recognized by both antibodies. To ensure that N or RBD proteins are recognized by both antibodies and T cells from humans, we used samples from COVID-19 convalescent and individuals that have been vaccinated with an inactivated virus vaccine (CoronaVac). The time of serum samples collection varied from 2 to 8 months after viral detection by RT-PCR in convalescents and 1 to 2 months after the second dose in vaccinated individuals. Convalescent individuals developed a high but variable antibody response to S (Supplementary Fig. 1d) and N proteins (Supplementary Fig. 1e) and a low response to the RBD expressed in bacteria (Supplementary Fig. 1f). Vaccinated individuals showed a low antibody response to both N and RBD proteins (Supplementary Fig. 1e, f), and a high response to the S protein (Supplementary Fig. 1d). Next, we evaluated the response of PBMCs to the S, N and RBD proteins. Importantly, PBMCs from convalescent and vaccinated individuals, but not seronegative healthy donors (HD), showed a robust IFN-γ response to the recombinant S (Supplementary Fig. 1g), N (Supplementary Fig. 2h) and RBD (Supplementary Fig. 1i) indicating that they are highly immunogenic for T cells.

We also evaluated the T cell response of immunized and convalescent individuals to SpiN using a Th1/Th2/Th17 cytometric bead array. The in vitro stimulation of PBMCs with SpiN induced the production of very high levels of IFN-γ and also IL-2, TNF, IL-6, and IL-10, but no IL-4 and IL-17, indicating that both vaccinated and convalescent individuals mounted a Type I helper T cell (Th1) response to the RBD and N proteins from SARS-CoV-2 (Supplementary Fig. 1j). We also evaluated which T cell population was producing IFN-γ, the main

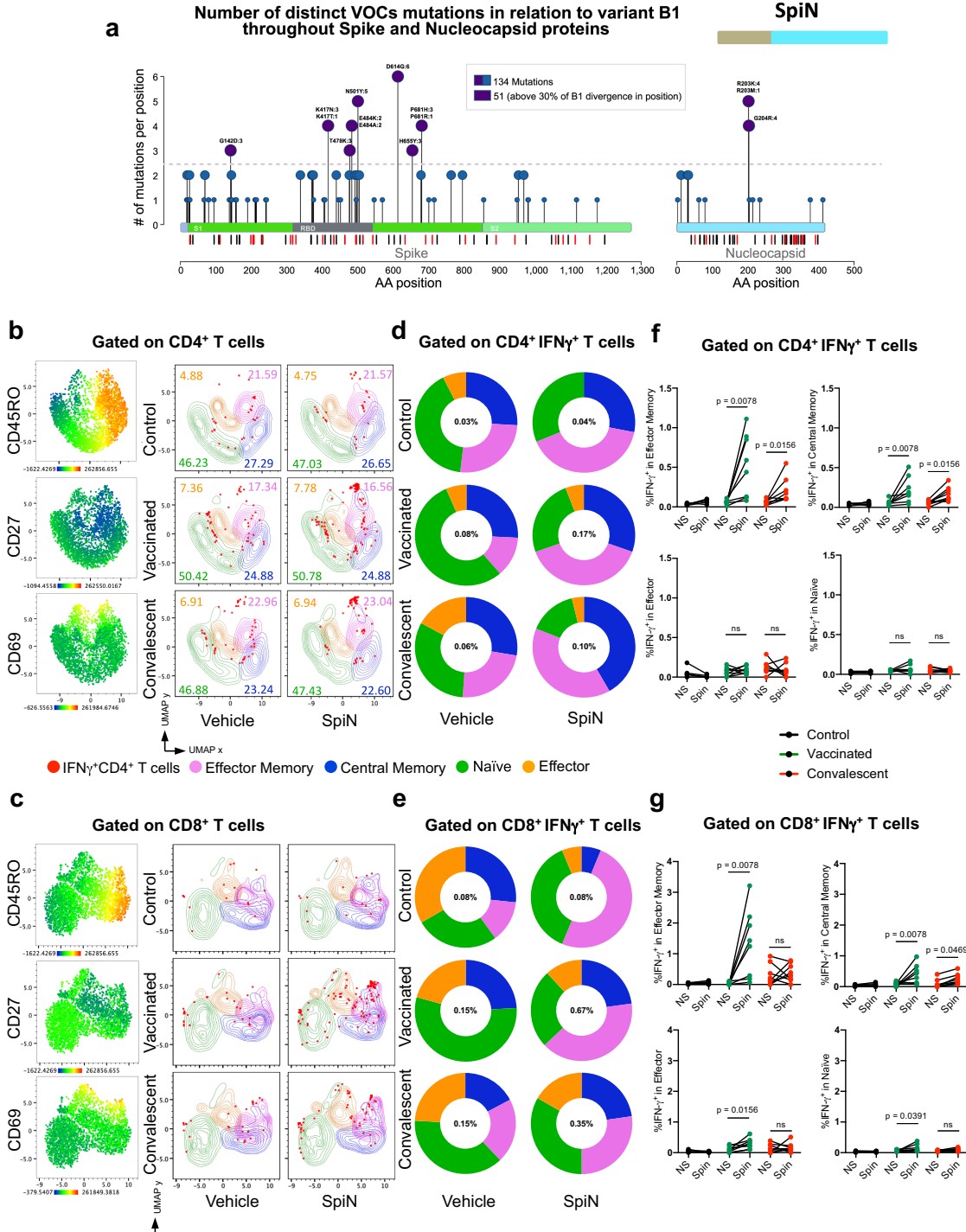

**Fig. 1 | Human antibody and IFNγ responses to N and RBD polypeptides.**
**a** Needle plot indicating the number of amino acid divergence points in the protein sequence of Nucleocapsid (N) and Spike (S) of five variants in relation to the N and S sequences from the SARS-CoV-2 lineage B (Wuhan). The peaks with circles indicate the position of the most frequent amino acid changes. The height of the peaks indicates the frequency of the changes in each divergent point. The blue and purple circles indicate common mutations and those observed on variants of concern, respectively. The sum of the amino acid changes for each segment (S1, RBD, and S2) of the S and N proteins considering the 6 SARS-CoV2 lineages is also shown. The vertical black and red lines below the bars illustrating the N and S polypeptides indicate, respectively, the position of each putative CD4+ T and CD8+ T cell epitopes identified by in silico epitope prediction. **b-g** UMAP projection of FACS data showing IFN-γ production by different CD4+ (**b**) or CD8+ (**c**) T cell compartments, determined by the surface markers CD45RO, CD27, and CD69. Data are also represented by the percentage of each subpopulation in total CD4+ (**d, f**) or CD8+ (**e, g**) IFN-γ+ T cells. The number of individuals used in these experiments was 5 controls, 8 vaccinated, and 8 convalescents. Statistical analysis of IFN-γ production (**f, g**) was performed using two-sided Wilcoxon-matched pairs signed rank; "ns" indicate that difference is not statistically significant and NS = non-stimulated PBMCs.

cytokine produced by Th1 lymphocytes. The gating strategy is shown in Supplementary Fig. 2. First, we gated on CD4[+] T cells (Fig. 1b) or CD8[+] T cells (Fig. 1c). The cell surface markers CD45RO, CD27, and CD69 were used to define the central memory, effector memory as well as the effector and naïve T cells. Next, we evaluated which are the main T cell subsets expressing IFN-γ. Both IFNγ-producing central and effector memory CD4[+] T and CD8[+] T cells expanded, whereas the effectors and naïve subsets contracted or remained unchanged after stimulation with SpiN (Fig. 1b–e). We also found that among both CD4[+] T (Fig. 1d) and CD8[+] T lymphocytes (Fig. 1e), the main subset expressing IFN-γ were the effector memory T cells, followed by the central memory T cells. The frequency of IFNγ-producing central memory, effector memory, effector T cells and naïve CD4[+] and CD8[+] T lymphocytes, before and after stimulation with SpiN, are shown in Fig. 1f, g, respectively.

## Antibody response to SARS-CoV-2 antigens in mice immunized with SpiN

Next, we evaluated the immunogenicity and whether immunization with RBD, N, or SpiN protects mice against the SARS-CoV-2 challenge. Mice were immunized with either recombinant protein by giving two intramuscular doses scheduled 21 days apart (Fig. 2a). As an immunological adjuvant we used the synthetic polyinosinic-polycytidylic acid (Poly I:C) mixed with the stabilizers carboxymethylcellulose and polylysine (Poly-ICLC, Hiltonol). We have chosen this adjuvant because it has been shown to induce an effective immunity to influenza that also infects humans through the respiratory tract[38,39]. In addition, the Poly-IC derivatives are potent activators of Toll-Like Receptor 3 (TLR3) and MDA5 from RIG-I family. Both cytosolic innate immune receptors also recognize double-stranded RNA and favors T cell-mediated immunity[40,41]. It is noteworthy that Poly ICLC has also been used in multiple clinical trials for cancer therapy[40,42,43].

Our results demonstrate that either recombinant RBD or N protein associated with Poly ICLC are immunogenic inducing, respectively, high levels of anti-RBD or anti-N antibodies, both in the bronchioalveolar fluid (BALF) (Fig. 2b) and sera (Fig. 2c) of vaccinated mice. We also observed a strong IFN-γ (Fig. 2d) and IL-10 (Fig. 2e) response by splenocytes stimulated with either N or RBD. The K18-ACE-2 mice are a model of severe disease[44], and were used to evaluate the efficacy of immunization with N or RBD recombinant proteins associated with Poly ICLC. Our results show that immunization with either protein resulted in partial protection to SARS-CoV-2 challenge, as indicated by body weight loss (Fig. 2f), mortality (Fig. 2g) as well as the viral load in the lungs (Fig. 2h) and brain (Fig. 2i).

Importantly, we report that immunization with adjuvanted SpiN induced robust viral-specific T cell and antibody responses and is highly efficacious in protecting against experimental challenge with the SARS-CoV-2. Sera from mice immunized with SpiN associated to Poly ICLC showed very high titers of IgG antibodies specific for RBD (1:5,000) and N proteins (1:25,000) (Fig. 2j) as well as inactivated virus (1:5,000), but low titers of anti-S antibodies (Fig. 2k). The levels of IgG anti-N (1:400) (Fig. 2l) as well as anti-SARS-CoV-2 (1:200) (Fig. 2m), were higher in sera from COVID-19 convalescent individuals than from healthy controls, but relatively low when compared to SpiN-immunized mice (Fig. 2j, k). In contrast to immunized mice, the titers of anti-RBD were low (Fig. 2l) and anti-S (1:200) high (Fig. 2m) in sera of convalescent individuals. The results presented in Fig. 2n show the increased levels of antibodies anti-N (**left panel**) and anti-RBD (**right panel**) in the BALF of mice vaccinated with SpiN.

Consistent with the high expression of the N protein[37] in infected cells, antibodies from mice immunized with either N or SpiN proteins strongly recognized UV-inactivated SARS-CoV-2 in an ELISA (Fig. 2o). Relevant to this study, we found that immunization with neither N, RBD nor SpiN chimeric protein elicited nAbs to the SARS-CoV-2, contrasting with the measurable levels in sera of COVID-19 convalescent

patients (Fig. 2p). In agreement with the ELISA results (Fig. 2o), sera from mice immunized with the N protein strongly reacted with paraformaldehyde fixed SARS-CoV-2 infected cells showing a diffuse expression of N protein in the cytosol, as demonstrated by immunofluorescence (IFA) (Fig. 2q). In contrast, the antibodies from mice immunized with RBD reacted with small vesicles, which might be consistent with Spike protein assembly in the endoplasmic reticulum–Golgi intermediate compartment (ERGIC), before proper folding[45]. The reactivity of sera from SpiN-immunized mice showed a mixed pattern consistent with a diffuse expression of N protein in the cytosol and a punctate staining of RBD, as observed with anti-N and anti-RBD antisera, respectively (Fig. 2q).

## T cell response to SARS-CoV-2 antigens in mice immunized with SpiN

In order to evaluate the polarization of the immune response induced by vaccination, we measured the levels of cytokines in the supernatants from splenocytes stimulated with RBD or N protein. Our results show that high levels of IFN-γ, and in a less extent IL-6 and IL-10, were produced by splenocytes from SpiN-vaccinated mice stimulated with either RBD (Fig. 3a) or N (Fig. 3b) proteins. In contrast, the levels of IL-4 and IL-17 were either low or not produced in response to either RBD or N antigens (Fig. 3a, b). In mice immunized with SpiN associated to Poly ICLC, the levels of IgG2c antibodies to RBD are higher than IgG1, showing a trend to a type 1 immune response (Fig. 3c). In contrast, there was no difference in the levels of IgG1 and IgG2c antibodies specific for the N protein as well as inactivated SARS-CoV-2 (Fig. 3d, e). As shown for total IgG (Fig. 2k), the levels of IgG1 and IgG2c antibodies specific to the S protein were very low (Fig. 3f). We also evaluated the recall T cell response in vaccinated mice. Both the N and RBD proteins induced high levels of CD4[+]CD44[+] and CD8[+]CD44[+] activated T cells from spleens of SpiN-vaccinated, but not control mice, as indicated by flow cytometry (Fig. 4a, b). To generate the UMAP, we gated on CD4[+] T cells (Fig. 4c, **left panels**) or CD8[+] T cells (Fig. 4c, **right panels**). The cell surface markers CD44 and CD62L, were used to define the central memory (Tcm), effector/effector memory (Teff/em) and naïve T cells. The cells were also stained with anti-IFN-γ. We found that CD4 Teff/em and CD8 Tcm lymphocytes were the main sources of IFN-γ in splenocytes from immunized mice (Fig. 4d, e, Supplementary Fig. 3a).

We also evaluated the local T cell response in the lungs by flow cytometry (Fig. 4f, g, Supplementary Fig. 3b), and it was observed that SpiN-immunized mice present an increase of tissue-resident memory (Trm) CD4[+] (Fig. 4f, **left panel**) and CD8[+] (Fig. 4g, **left panel**) T cells when stimulated with SpiN. Importantly, SpiN also induced the production of IFN-γ and TNF by both CD4[+] and CD8[+] Trm from SpiN-immunized mice (Fig. 4f, g, **middle and right panels**). Hence, it is likely that local T cell responses have an important function in controlling viral growth and consequent inflammation in SpiN-immunized mice.

## Immunization with SpiN limits SARS-CoV-2 replication and protects against disease

The Syrian hamsters were used in our experiments as a model of moderate COVID-19 disease[46]. The results presented in Fig. 5a show that SpiN associated with Poly ICLC induced high levels of total IgG anti-N in immunized hamsters. The levels of anti-RBD were lower (Fig. 5a). Of note, the levels of nAbs in vaccinated hamsters were undetectable (Fig. 5b). Nevertheless, the viral load detected by RT-PCR was significantly lower in vaccinated as compared to unvaccinated hamsters challenged with SARS-CoV-2 (Fig. 5c). Histopathological analysis of hamsters at 4 days post-infection (dpi) with the Wuhan strain shows that the lungs in the PBS group display a diffuse interstitial pneumonia (Fig. 5d, **right panels**). In comparison, hamsters immunized with SpiN adjuvanted with Poly ICLC had mild focal congestion (black arrow) with alveolar space preservation (Fig. 5d, **left panels**).

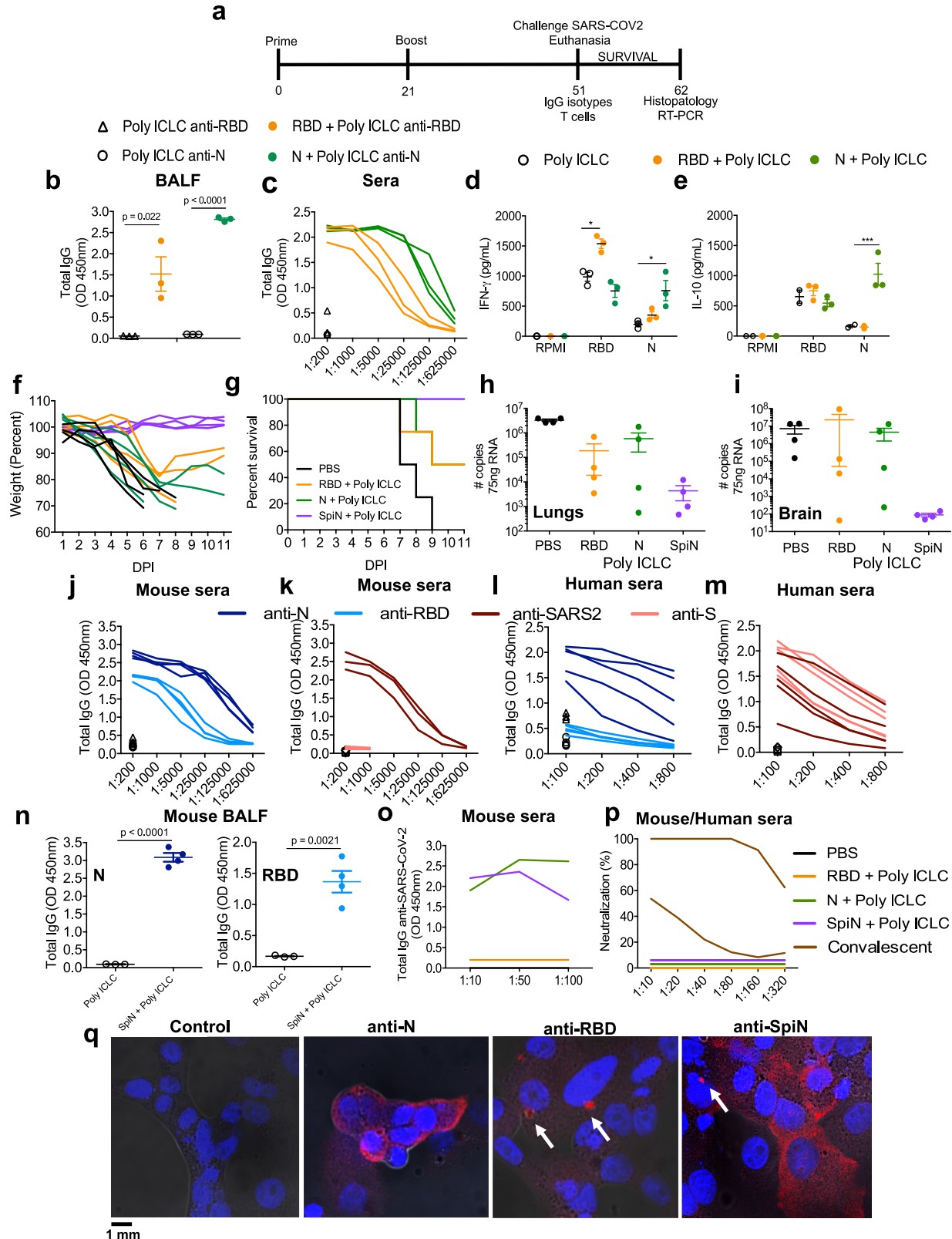

Immunization with SpiN associated to Poly ICLC also protected the K18-ACE-2 mice, a model of severe COVID-19, from weight loss (Fig. 6a) and other clinical signs of disease, such as affected motility, ruffle fur, and hunching. Importantly, 100% of immunized mice survived infection, whereas all mice that received Poly ICLC alone succumbed to infection (Fig. 6b). In addition, at 5 dpi the viral RNA load both in the lungs and brain (Fig. 6c, d) were lower, when comparing SpiN-vaccinated versus non-vaccinated controls that received adjuvant alone, as indicated by RT-PCR. Importantly, the levels of nAbs were not detectable in sera of mice immunized with SpiN at 5 dpi (Fig. 6e). Histopathological analysis demonstrates that the lungs from Poly ICLC group, at 5 dpi, showed a diffuse interstitial pneumonia (Fig. 6f, **top panels**). In comparison, the immunized group (SpiN plus Poly ICLC) showed

**Fig. 2 | Evaluation of immune response and protection elicited by vaccination with RBD, N or SpiN proteins associated with Poly ICLC. a** Immunization protocol used in the experiments shown in Figs. 2–8 to analyze the immune response and protection against SARS-CoV-2. Antigen-specific IgG antibodies were measured in the bronchoalveolar lavage (BALF) at 1:1 (**b**) and serially diluted sera from immunized mice (**c**). Levels of IFNγ (**d**) and IL-10 (**e**) were measured on culture supernatant of splenocytes stimulated with RBD or N antigens. **b-e**, *n* = 3/group. **f, g** Body weight and survival of K18-hACE2 mice immunized with RBD, N or SpiN associated with Poly ICLC and challenged with the Wuhan strain of SARS-CoV-2. **h, i** Viral load, measured by RT-PCR, in the lung and brain tissues from unvaccinated and mice vaccinated with either RBD, N, or SpiN associated with Poly ICLC. **f-i** *n* = 4/group. **j, k** Levels of circulating total IgG specific for either N, RBD and S proteins or inactivated SARS-CoV-2 in mice immunized with SpiN (*n* = 4/group). **l, m** Total IgG response of convalescent individuals to either N or RBD and inactivated SARS-CoV-

2 or S (n = 5/group), respectively. **n** Levels of anti-N (**left panel**) and anti-RBD (**right panel**) antibodies were measured in BALF from mice that received Poly ICLC alone or SpiN plus Poly ICLC (n = 4/group). **j–n** = 3-5/group. Levels of anti-inactivated SARS-CoV-2 (**o**) and neutralizing antibodies (**p**) in pooled sera from mice immunized with PBS, N, RBD or SpiN and from COVID-19 convalescent individuals. **q** Immunofluorescence of SARS-CoV-2 infected cells stained with sera from mice that received adjuvant alone (**left panel**) or were immunized with N (**left middle panel**), RBD (**right middle panel**) or SpiN (**right panel**) associated with Poly ICLC. Arrows are pointing to small vesicles containing the S protein that reacted with anti-RBD and anti-SpiN polyclonal antibodies. Statistical analysis of IgG measured in BALF (**b, n**) was performed using two-tailed unpaired *t*-test. Cytokine measurements (**d, e**) were analyzed through Two-way ANOVA followed by Tukey's multiple tests. Data are representative of two independent experiments. **b, d–e, h–i, n**, data are presented as mean values ± SEM. * *P* < 0.05 and *** *P* < 0.001.

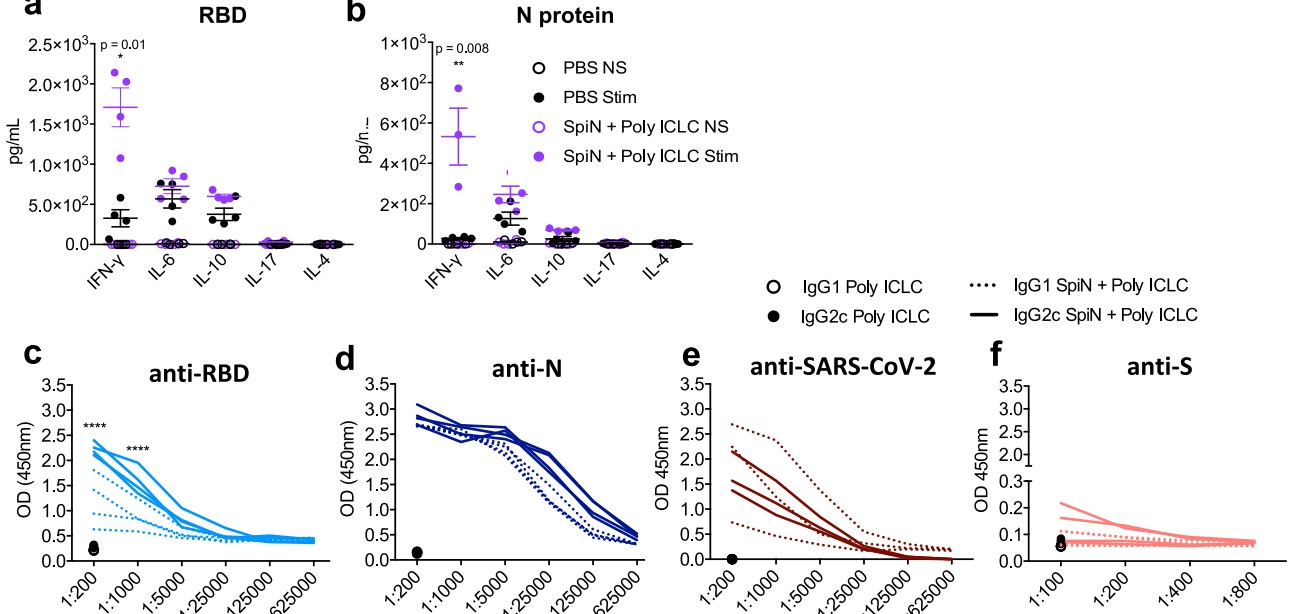

**Fig. 3 | Levels of cytokines and anti-SARS-CoV-2 IgG isotypes from mice immunized with SpiN.** Levels of cytokines in the supernatant of splenocytes from control and vaccinated mice stimulated (stim) or not (NS) with the RBD (**a**) and N proteins (**b**), as measured by the cytometric bead array (CBA). The levels of IgG1 and IgG2c specific for RBD (**c**), N (**d**), SARS-CoV-2 (**e**) and S (**f**) proteins in sera from

mice that received adjuvant alone (black symbols in the bottom) or were immunized with SpiN protein associated with Poly ICLC. Data are representative of two independent experiments, *n* = 3–4 mice/group. Statistical analysis of CBA was performed with multiple *t* tests, and antibody measurements were analyzed using Two-way ANOVA followed by Sidak's multiple comparisons test. **** *P* < 0.0001.

preservation of the pulmonary architecture (Fig. 6f, **bottom panels**).

Consistent with the intense inflammatory response, we found high mRNA levels of IL-6 and TNF (Fig. 6g), as well as chemokines (CCL2, CCL5, CXCL9, and CXCL10) (Fig. 6h), in the lungs from non-vaccinated mice challenged with SARS-CoV-2. In contrast, the level of type 2 cytokines (IL-4 and IL-5) was higher in the vaccinated mice (Fig. 6g), suggesting that the microenvironment in the lung mucosal environment was preserved in vaccinated mice, whereas a dramatic switch to an inflammatory response was observed in the lungs of mice that developed severe disease.

The frequency of total myeloid cells was dramatically increased (Fig. 6i) and lymphoid cells decreased (Fig. 6j) in the lungs of non-vaccinated mice challenged with SARS-CoV-2, whereas immunization prevented the intense inflammatory infiltrate. Consistently, the total number of neutrophils (Fig. 6k), monocytes (Fig. 6l), and monocyte-derived dendritic cells (Fig. 6m) were dramatically increased in the lungs of non-vaccinated mice infected with SARS-CoV-2. In contrast, the numbers of resident CD8+ T cells (Fig. 6n) and conventional DCs (Fig. 6o) were increased in vaccinated mice, further suggesting a

possible involvement of the resident T cells in resistance to SARS-CoV-2 and COVID-19[47]. Thus, SpiN associated with Poly ICLC was highly immunogenic and induced robust protection against SARS-CoV-2.

## SpiN-induced protective immunity to SARS-CoV-2 is primarily mediated by T lymphocytes and not nAbs

A relevant finding of this study is the demonstration that mice immunized with SpiN are highly resistant to SARS-CoV-2, even in absence of circulating nAbs at the time of infection. An important question is regarding the levels of nAbs in immunized rodents soon after the challenge with SARS-CoV-2. We found that in vaccinated-hamsters at 4 dpi (Fig. 5b) and mice at 5 dpi (Fig. 6e), the levels of nAbs remained undetectable, whereas inflammation was mild (Figs. 5d, 6f) and viral load in the lungs decreased (Figs. 5c and 6e). These results support the hypothesis of a nAb-independent immunity in mice immunized with SpiN.

To further investigate whether CD4+ and CD8+ T cells are essential for controlling of viral load and protective immunity against SARS-CoV-2, SpiN-vaccinated mice were treated with either or both anti-CD4 and anti-CD8 antibodies, at days −3, −2, and −1 before the challenge. T

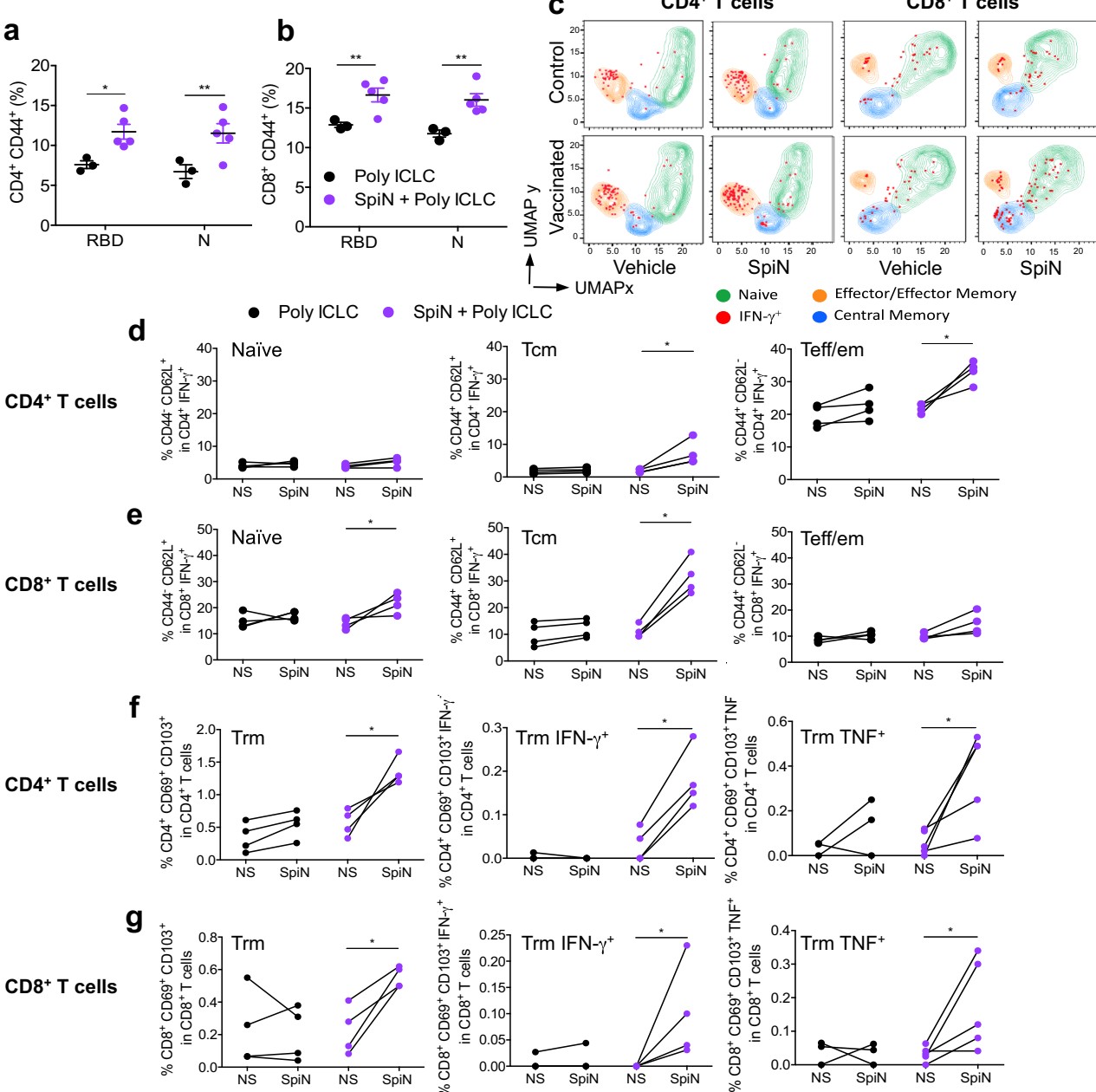

**Fig. 4 | T lymphocytes response in the spleen and lungs of SpiN-immunized mice.** The frequency of activated CD4⁺ T (**a**) and CD8⁺ T (**b**) lymphocytes was evaluated by measuring the expression of cell surface CD44, data are presented as mea ± SEM. **c** UMAP projection of FACS data, in which splenocytes were gated on either CD4⁺ T or CD8⁺ T cell populations. Staining with anti-CD62L, anti-CD44 and anti-IFN-γ was used to define IFNγ-producing naïve, central memory (Tcm), and effector/effector memory (Teff/em) T cells. The frequency of naïve, memory and effector CD4⁺ T and CD8⁺ T populations producing IFN-γ with or without stimulation with SpiN are shown in panels **d** and **e** respectively. **f, g** Imunophenotyping of the lungs from mice immunized or not. The cells were cultured with SpiN protein and characterized by flow cytometry as CD4⁺ CD69⁺ CD103⁺ (**f, left panel**), CD4⁺ CD69⁺ CD103⁺ IFN-γ⁺ (**f, middle panel**), CD4⁺ CD69⁺ CD103⁺ TNF⁺ (**f, right panel**), CD8⁺ CD69⁺ CD103⁺ (**g, left panel**), CD8⁺ CD69⁺ CD103⁺ IFN-γ⁺ (**g, middle panel**) and CD8⁺ CD69⁺ CD103⁺ TNF⁺ (**g, right panel**). Data are representative of two independent experiments, n = 3–4 mice/group. Statistical analysis was performed using two-sided Mann–Whitney. * P < 0.05 and ** P < 0.01.

cell depletion of over 90% was confirmed by flow cytometry (Supplementary Fig. 4). As depicted on Fig. 7a, b, 50% of mice depleted of either CD4⁺ or CD8⁺ T cells showed significant weight loss and succumbed to infection, while 100% of mice depleted of both CD4⁺ and CD8⁺ T lymphocytes lost weight and died up to 8 days post-infection. Altogether, our results indicate that protection in mice immunized with SpiN, at an early stage of SARS-CoV-2 infection, is mediated by effector T cells and not nAbs.

Although vaccinated mice produced extremely high levels of anti-N antibodies, we have no evidence that they mediate resistance to

SARS-CoV-2. Since the N protein associates with the viral RNA genome and is not expressed in the viral surface membrane, it is unlikely that anti-N antibodies act by mediating antibody dependent cell cytotoxicity (ADCC), by promoting phagocytosis of the opsonized virus, or by blocking host cell invasion by SARS-CoV-2. Consistently, our results show that anti-N antibodies are uncapable of blocking in vitro invasion of the host cell by SARS-CoV-2 (Fig. 2p). In order to evaluate if SpiN-induced antibodies are able to protect mice against SARS-CoV-2 infection, we transferred serum from immunized mice to naïve K18-ACE-2 mice at day −1 before the challenge. All animals that received

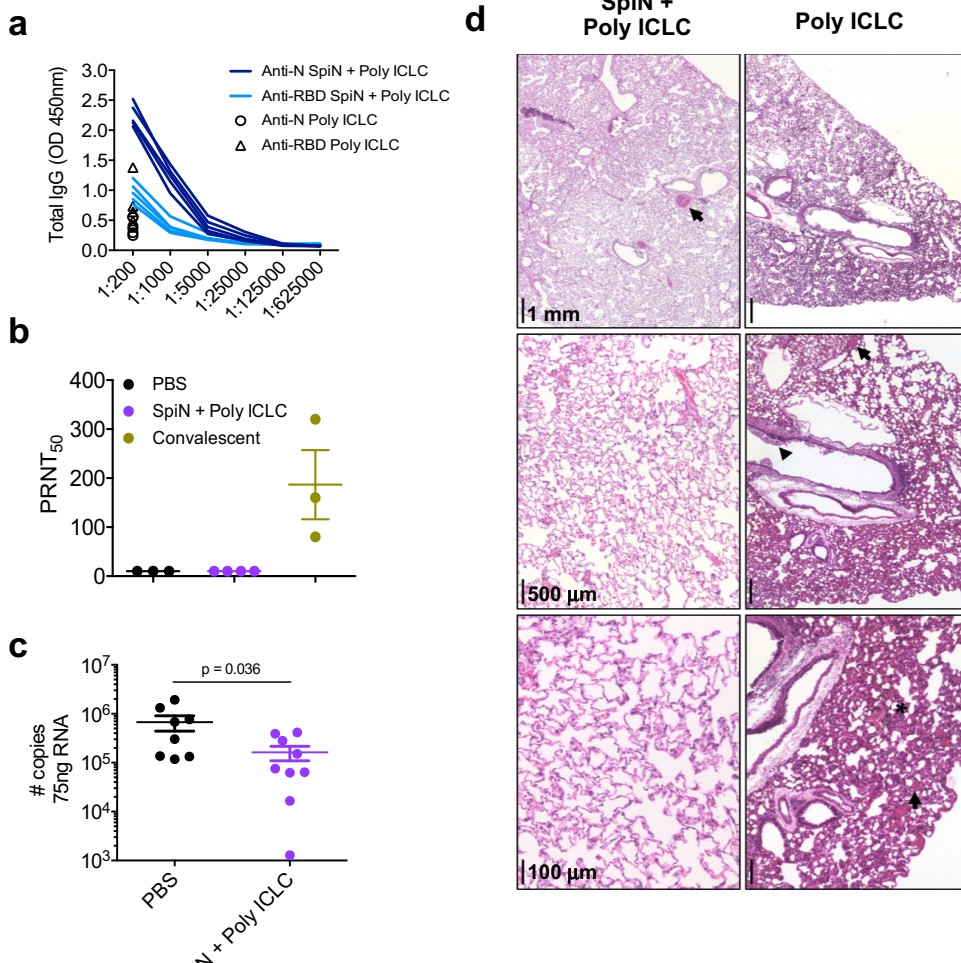

**Fig. 5 | Protective immunity in hamsters immunized with SpiN and challenged with SARS-CoV-2. a** The levels of total IgG anti-N and anti-RBD in sera of hamsters vaccinated with SpiN associated to Poly ICLC. **b** Levels of neutralizing antibodies in the sera from hamsters vaccinated with SpiN + Poly ICLC, in comparison with sera from convalescent individuals. **c** Viral load measured by RT-PCR in the lungs of control and hamsters immunized with SpiN plus Poly ICLC at 4 days post-infection (dpi) with SARS-CoV-2. Data are presented as mean +/- SEM (**b, c**).
**d** Histopathological analysis at 2,5x, 5x, and 20x magnification of hamsters at 4 dpi with the Wuhan strain, shows that the lungs in the PBS group display an accentuated diffuse alveolar wall thickening, with moderate multifocal collapse

(asterisk), associated with accentuated diffuse congestion (black arrow) and mixed inflammatory infiltrate (mononuclear and polymorphonuclear cells). In the bronchial space, inflammatory cells are noted with a predominance of neutrophils associated with cellular debris (arrowhead) (**d, right panels**). In comparison, hamsters immunized with SpiN adjuvanted with Poly ICLC had mild focal congestion (black arrow) with alveolar space preservation (**d, left panels**). **a–d** Data are representative of two independent experiments. **a** n = 5 hamsters/group. **d** PBS n = 3, SpiN + Poly ICLC n = 4, Convalescent n = 3. **c** Pooled data from two independent experiments, n = 7–8 hamsters/group. Data of viral load quantification was analyzed through two-sided Mann–Whitney test.

serum from either control or vaccinated mice presented body weight loss and succumbed to SARS-CoV-2 infection (Fig. 7c, d).

As control, we evaluated the effect of T cell depletion as well as the transfer of sera from mice vaccinated with an adenoviral vector encoding the S protein (Covishield). The immunized mice show high levels of neutralizing antibodies (PRTN > 320). In contrast to SpiN-vaccinated mice, 100% of Covishield-vaccinated animals depleted of both CD4+ and CD8+ T cells survived (Fig. 7e, f) challenge with the Wuhan isolate. Consistent with the main function of nAbs in this model, transfer of sera from Covishield-immunized mice protected the naïve K18-ACE-2 mice from challenge with SARS-CoV-2 (Fig. 7g, h).

### Immunization with SpiN protects against the Delta and Omicron variants
Finally, the Delta and the Omicron have emerged in 2021 as VOCs with higher infectivity[48,49] that efficiently escape recognition by nAbs, both in vitro and in experimental models[50–54]. Furthermore, the efficacy of current S-based vaccines used for mass vaccination have lower efficacy

against infection with either Delta or Omicron isolates[55–61]. Hence, we evaluated whether immunization with SpiN protects mice against challenges with these VOCs. Our results show that immunization with SpiN fully protected the K18-ACE-2 mice against infection with the Delta variant as measured by loss of body weight and lethality (Fig. 8a, b). Consistent with a previous study, naïve K-18-ACE-2 mice did not lose weight or succumbed to infection with Omicron[51] (Fig. 8c). Nevertheless, immunization with SpiN significantly protected the K18-ACE-2 mice from challenge with Omicron isolate, as indicated by a 16 fold decreased in viral load measured by RT-PCR at 6 dpi (Fig. 8d). Furthermore, in control mice that received Poly ICLC only, the lungs showed a diffuse interstitial pneumonia (Fig. 8e, **top panels**), contrasting with preserved pulmonary architecture in SpiN-immunized mice (Fig. 8e, **bottom panels**).

## Discussion
Despite of the substantial loss of nAbs the current COVID-19 vaccines retain the ability to induce a significant level of protection against

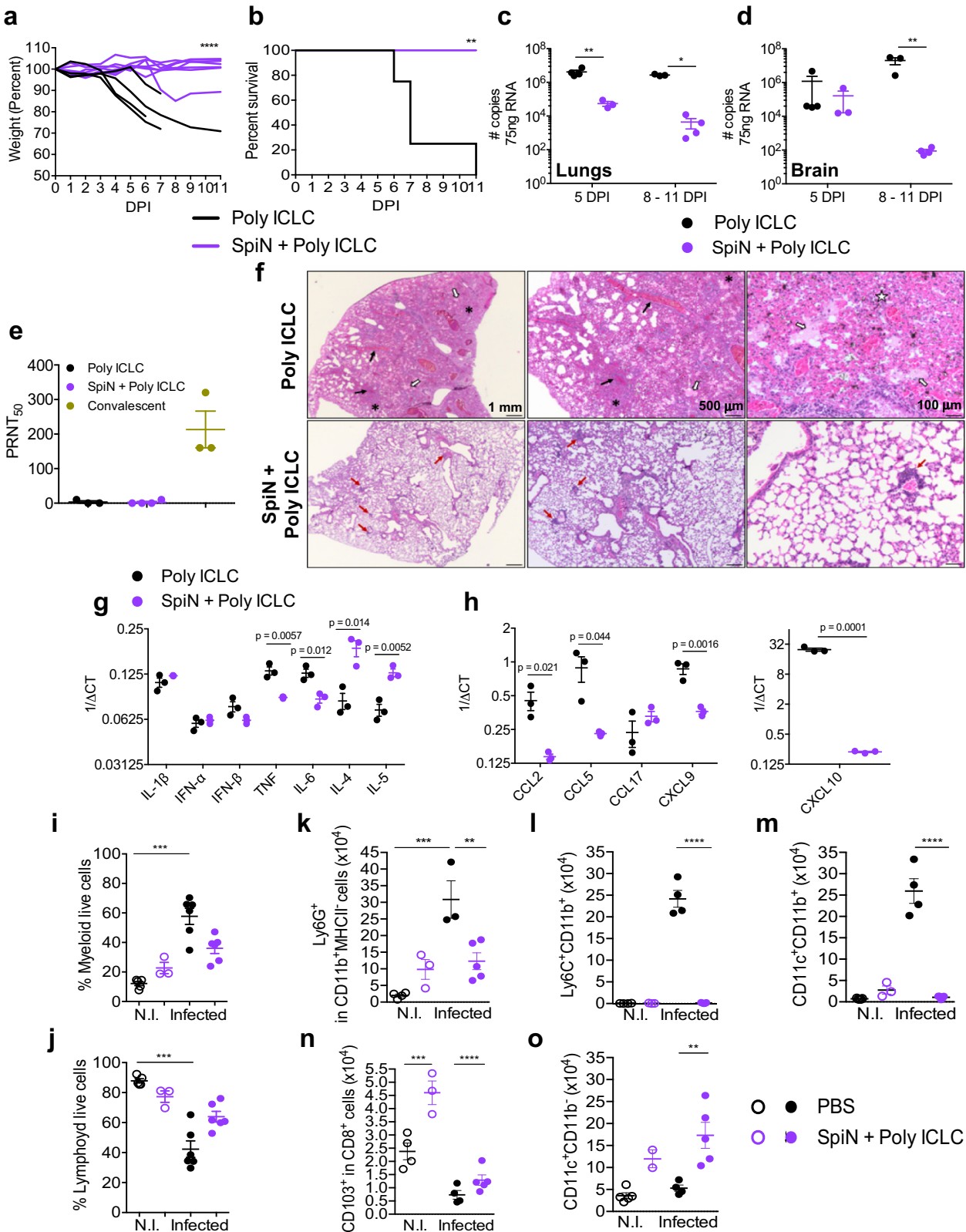

disease. In fact, most of T cell responses are preserved against the VOCs and are likely responsible, at least in part, for their protective effect[62,63]. Yet, the S protein of the Omicron variant has over 30 mutations, and the loss of T cell responses induced by vaccination is estimated to be around 30%[64], further undermining protective immunity. Hence, the use of antigens, like the N protein, that is highly enriched for conserved T cell epitopes[29], resistant to selection of non-

synonymous mutations (Fig. 1a and Supplementary Tables 1–4), might be a good strategy towards a universal vaccine against the SARS-CoV-2 variants.

A limited number of studies have tested peptide-based COVID-19 vaccines for immunogenicity, both in mice and humans[65,66]. Remarkably, a vaccine containing multiple T cell epitopes, shared by SARS-CoV-2 variants, was tested in a human phase I clinical trial showing

**Fig. 6 | Protective immunity in mice immunized with the SpiN chimeric protein and challenged SARS-CoV-2.** Mice were challenged with the Wuhan strain of SARS-CoV-2. Body weight (**a**), survival (**b**) and viral load in lungs (**c**) and brain (**d**) were evaluated in K18-hACE-2 mice immunized with SpiN plus Poly ICLC and controls that received Poly ICLC alone. **e** Titers of nAbs in the sera from control and immunized mice at 5 dpi. Convalescent patients were used as positive control. **c**–**e** Data are presented as mean ± SEM. **f** Histopathological analysis at 2,5x, 5x, and 20x magnification of the lungs from Poly ICLC or SpiN + Poly ICLC groups, at 5 dpi. Black arrows: congestion; white arrows: intra-alveolar exudate; white star: hemorrhagic foci; asterisks: alveolar collapse; red arrows: inflammatory infiltrate. **g**, **h** mRNA expression of cytokines (**g**) and chemokines (**h**) quantified by qRT-PCR in mice lungs at 5 dpi. **i**–**o**, Frequency of myeloid (**i**) and lymphoid (**j**) as well as total numbers of neutrophils (**k**), monocytes (**l**), monocyte derived dendritic cells (**m**),

resident CD8[+] T cells (**n**) and conventional dendritic cells (**o**) in the lungs of control, vaccinated, challenged (infected) or not (NI) with SARS-CoV2 at 5 dpi. **g**–**o** Data are presented as mean ± SEM. **a**, **b** Individual values of pooled data from two independent experiments, Poly ICLC *n* = 4 and SpiN + Poly ICLC *n* = 7. **c**–**o** Data are representative of two independent experiments, *n* = 3 mice/group (**c**–**h**). **i**–**o**, *n* = 4 PBS N.I. and infected, *n* = 3 SpiN + Poly ICLC N.I., *n* = 6 SpiN + Poly ICLC infected. Statistical analysis of weight measurements (**a**) was performed using Two-way ANOVA. Survival analysis (**b**) was performed with Log-rank test. Data of viral quantification (**c**, **d**) was analyzed using Two-way ANOVA followed by Sidak's multiple comparisons test. qRT-PCR data (**g**, **h**) was analyzed with unpaired two-tailed t tests. Flow cytometry of the lungs (**i**–**o**) was analyzed by Kruskal–Wallis followed by Dunn's multiple comparisons test. * *P* < 0.05, ** *P* < 0.01, *** *P* < 0.001 and **** *P* < 0.0001.

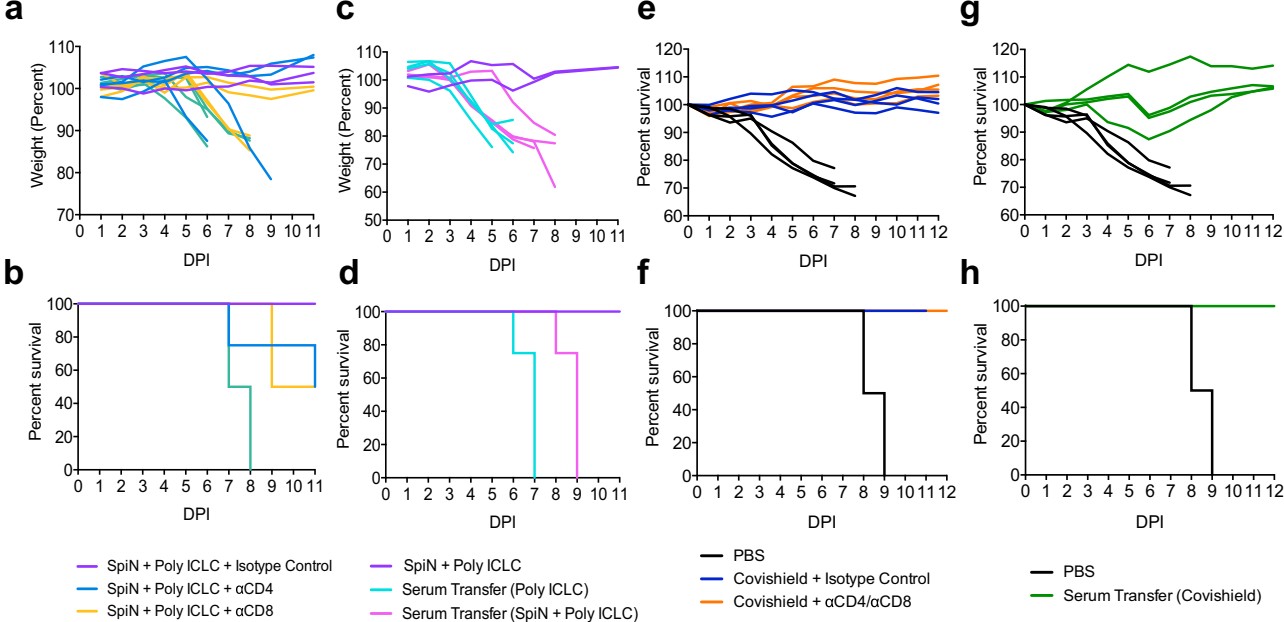

**Fig. 7 | The importance of T lymphocytes and antibodies in the protective immunity elicited by SpiN.** K18-hACE2 mice were immunized with SpiN adjuvanted with Poly ICLC (**a**, **b**) or Covishield (**e**, **f**) and treated with anti-CD4[+], anti-CD8[+] or both on days −3, −2, and −1 before infectious challenge. Naïve K18-hACE2

mice were administered, at day −1 before infection, with pooled sera from control or SpiN- (**c**, **d**) or Covishield- (**g**, **h**) immunized mice. Body weight (**a**, **c**, **e**, **g**) and survival (**b**, **d**, **f**, **h**) were monitored for 11 days after infection with the SARS-CoV-2 Wuhan strain. **a**–**h** 4 mice/group.

encouraging results in terms of T cell responses and safety[66]. However, the efficacy of these vaccines was not evaluated. The SARS-CoV-2 N protein seems an excellent alternative for a peptide vaccine since this antigen is highly enriched for putative CD4[+] and CD8[+] T cell epitopes across the main HLA haplotypes. In addition, children with asymptomatic or mild infections have an enhanced CD8[+] T cell response to the N antigen[35]. In addition, the SpiN vaccine, which should be very safe to be injected in humans, could be preferentially used in individuals with morbidities that affect the antibody responses as well as for children.

Other studies have taken in consideration the use of the SARS-CoV-2 N protein as a candidate for a COVID-19 vaccine[67–70]. However, they have either been associated with the S or RBD proteins that elicit nAbs[68,69] or used the N protein alone[70], which similar to our findings (Fig. 2f–i) elicited partial protection. Furthermore, in these studies, the function of antibodies versus T cells was not explored in detail and protection against a variant was not evaluated. Here, we show that the inclusion of the RBD region in our chimeric protein was necessary to protect 100% of the mice challenged with the wild type as well as a variant of SARS-CoV2. In addition, we show that the N/RBD fusion protein elicits a robust immunity to the Delta and Omicron variants.

Considering the number of potential T cell epitopes found in the N protein added to those found in the RBD region of the S protein, we assume that immunization with SpiN would elicit a diverse repertoire of T cells. This is central to T-cell mediated immunity to viral infection and highly relevant to overcome the SARS-CoV-2 plasticity. Since mutations are generated stochastically, it is unlikely that they will occur in various T cell epitopes simultaneously. Moreover, the selective pressure in one or few specific mutations is ineffective in undermining protective immunity that relies in a T cell response that targets multiple epitopes. Hence, we hypothesize that a vaccine containing a more broad array of linear T cell epitopes that promote both CD4[+] helper T and CD8[+] cytotoxic T cell responses would also be more effective against SARS-CoV-2 variants that evade the protective nAbs[4,6,7,9,10].

In conclusion, we report that immunization with N and RBD fusion protein (SpiN) expressed in a prokaryote system, adjuvanted with Poly ICLC, was highly immunogenic for T cells. This vaccine is cost-effective, stable, safe and very efficacious in protecting rodent models of COVID-19 against experimental challenges with either the wild type, the Delta or the Omicron variants of SARS-CoV-2. Altogether our results support the hypothesis that this protection is primarily

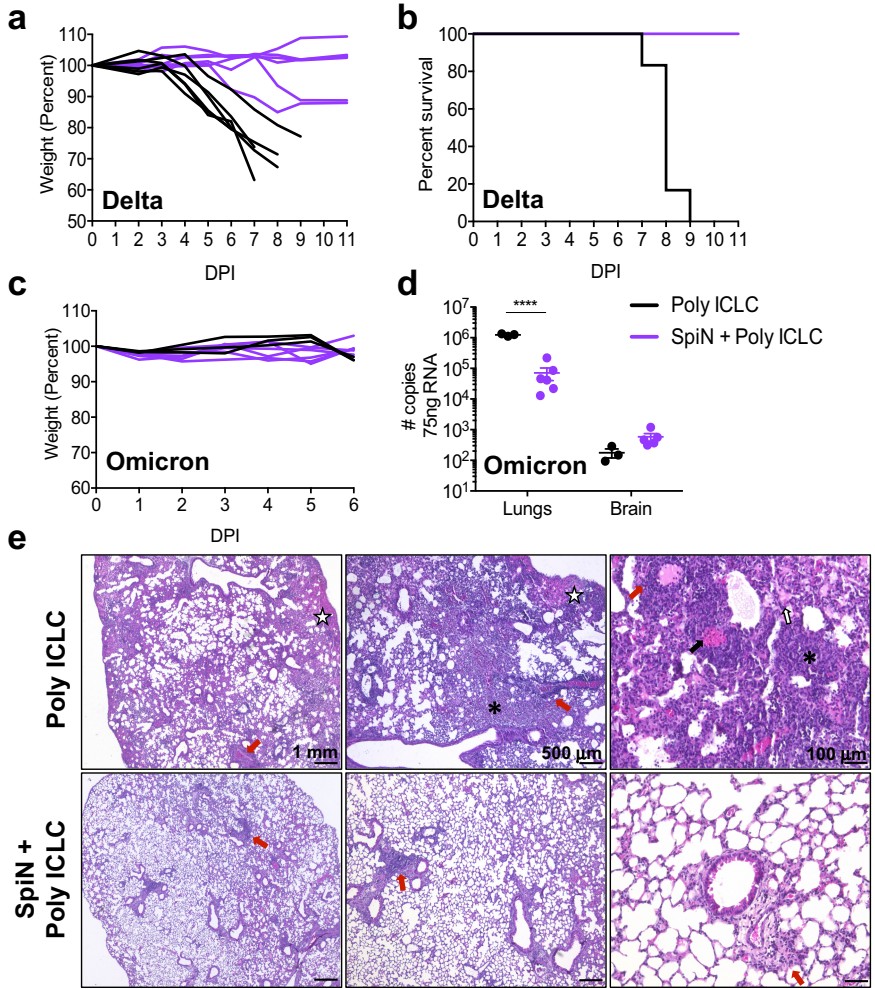

**Fig. 8 | SpiN-induced protective immunity to the Delta and Omicron variants of SARS-CoV-2.** K18-hACE2 mice were vaccinated with SpiN associated to Poly ICLC and challenged 30 days after the second dose with $5 \times 10^4$ PFU of Delta (**a**, **b**) or $2.5 \times 10^4$ PFU of Omicron (**c**, **d**) variants. Body weight (**a**, **c**) and survival (**b**) were measured for 11 days. **d** Viral load was measured by RT-PCR at 6 dpi with Omicron, data are presented as mean +/- SEM. **a**–**e** 6 mice/group. **e** Photomicrographs of lung tissue from mice infected with SARS-CoV-2 Omicron strain. Furthermore, in control mice that received Poly ICLC only, the lungs showed a diffuse interstitial pneumonia characterized by a mixed inflammatory infiltration (mononuclear and polymorphonuclear cells), accompanied by intense congestion (black arrows), intra-alveolar exudate (white arrows with black outline), hemorrhagic foci (white star) and areas of alveolar collapse (asterisks) (**e**, **top panels**). In the immunized group it is noted the preservation of the pulmonary architecture, with the presence of predominantly mononuclear inflammatory infiltrate (**e**, **bottom panels**). **a**–**e** Data are representative of two independent experiments. Statistical analysis of viral load (**d**) was performed with Two-way ANOVA followed by Sidak's multiple comparisons test. **** $P < 0.0001$.

mediated by CD4⁺ and CD8⁺ T cells and not nAbs. Although the definition of the specific sequences needs to be further investigated, this work suggests that SpiN is highly enriched for CD4⁺ and CD8⁺ T cell epitopes, and it is unlikely that non-silent point mutations will undermine SpiN-induced protective immunity. Hence, while not denying the importance of nAbs, the N protein and, more broadly, the use of multiple T cell epitopes should be considered to improve anti-COVID-19 vaccines to overcome SARS-CoV-2 plasticity.

## Methods

### Ethics statement

Ethical Committees on Human Experimentation from Fundação Hospitalar do Estado de Minas Gerais (FHEMIG) approved this study performed with human donors (CAAE: 43335821.4.0000.5119). Experiments with mice were conducted according to institutional guidelines for animal ethics and approved by the Institutional Ethics Committees from Oswaldo Cruz Foundation and Universidade Federal de São Paulo Ethics. Commission on Animal Use (CEUA) LW 25/20 and 105/2020, respectively.

### Blood donors

Human blood samples were collected from vaccinated individuals ($n = 33$), convalescent patients ($n = 13$), and healthy controls ($n = 9$). All individuals were between 18 and 70 years old ($36 \pm 11$, female:male ratio = 3.2) (Supplementary Table 5). Vaccinated individuals received two doses of Coronavac (Sinovac, China) and were sampled 27-54 days after the second dose. Convalescent individuals reported having mild COVID-19 between 24–196 days before sampling, confirmed by PCR. All individuals were briefly interviewed before sampling and consent forms were signed, and there was no participant compensation.

### Mice, hamsters and viruses

Female C57BL/6 mice, 6–10 weeks old, were purchased from the Center for Laboratory Animal Facilities of the Federal University of Minas Gerais (CEBIO-UFMG). Human Angiotensin Converting Enzyme transgenic mice (K18-hACE2) mice in the C57BL/6 background, 6–10 weeks old, originally from Jackson Laboratories, were bred at Fiocruz-Minas or at Fiocruz-São Paulo animal facilities and used as a model of severe COVID-19. Female Golden Syrian Hamsters,

8–10 weeks old were from Fiocruz-Minas Animal House and used as a model of mild COVID-19. The experiments were carried out following the recommendations of the Guide for the Care and Use of Laboratory Animals of the Brazilian National Council of Animal Experimentation (CONCEA). Mice and hamsters were bred and maintained in micro-isolators at Fiocruz-Minas and Universidade de São Paulo on a 12 h dark/light cycle, temperature range was 68–79 °F, and humidity between 30 and 70%. The severe acute respiratory syndrome coronavirus 2 (SARS-CoV-2) viral strain used in this study was from the lineage B (isolate BRA/SP02/2020), Delta (EPI_ISL_2965577) and Omicron (EPI_ISL_7699344) variants. Viral stocks were propagated in Vero E6 cells (ATCC CRL-1586) in a humidified incubator at 37 °C with 5% $CO_2$ and observed for cytopathic effects (CPE) daily up to 72 h. Viruses were titrated in Vero E6 cells by plaque forming units (PFU) assay[71]. Viral aliquots were kept at −80 °C until further use.

### Epitope prediction and sites of amino acid changes in the RBD region of the Spike (S) and Nucleocapsid (N) proteins from SARS-CoV-2

Epitope prediction was performed through The Immune Epitope Database and Analysis Resource (IEDB) platform for class I analysis and NetMHCII for class II[31,32]. Potential HLA-ABC, HLA-DR and mice MHC-I/II binding epitopes were sought in the sequences of Spike (6VSB https://doi.org/10.2210/pdb6VSB/pdb) and Nucleocapsid (7SD4_1 10.2210/pdb7SD4/pdb) proteins. We selected epitopes that presented binding scores <1 for HLA-ABC and HLA-DR; < 1.3 for mouse MHC I; and <2.0 for mouse MHC II. The Needle plot was constructed from the Spike and Nucleocapsid amino acid sequences recovered from the alignment of the reference genomes of the variants Alpha (MZ344997), Beta (MW598419.1), Delta (MZ359831.1), Gamma (MZ169911.1), Omicron BA.1 (OL672836.1), Omicron BA.2 (PRJNA784038). The coding sequences of the proteins were delimited, translated, realigned in order to identify the mutations acquired in relation to the original lineage (B1 – Wuhan – EPI_ISL_402123 https://www.epicov.org/epi3/frontend#4c5e54). Alignments were done using the MUSCLE tool, and default parameters. Mutations were counted and named by "homemade" scripts and the R mutsneedle package with specific modifications was used to construct the figure and incorporate information about the antigens.

### Plasmid constructions and recombinant antigens production

Plasmids containing sequences encoding the full length N, the RBD of Spike and the chimeric SpiN protein with codons optimized for expression in *E. coli* were purchased from Genscript. Competent *E. coli* Star (DE3) were transformed with the pET24 vector with N or the RBD sequences and *E. coli* pRARE with the pET24_with SpiN. Transformed bacteria were grown in LB medium with kanamycin (50 μg/ml) at 37°C until OD600 0.6 was reached. At this point, protein expression was induced by adding IPTG to the culture at a final concentration of 0.5 mM. The induction of expression of the three proteins was done at 37ºC for 3 h for N and RBB and for 18 h for SpiN. The N and RBD proteins contained a histidine tag and were purified through affinity chromatography step with the Histrap HP (GE HealthCare) column following the manufacturer's instructions. After bacterial lysis, N protein was purified from the soluble fraction and RBD protein from the insoluble fraction by adding 8 M urea in the buffers for solubilization. The SpiN protein, expressed without histidine tag, was purified from the soluble and insoluble fractions of the bacteria cell lisate after the addition of 8 M urea and through two steps of chromatography. A cation exchange chromatography with the Hitrap SP HP column (GE HealthCare) was followed by molecular exclusion with the column HiPrep 26/60 Sephacryl S-100 HR (GE HealthCare), following the manufacturer's instructions. The S protein expressed in mammalian cells was kindly provided by Dr. Leda Castilho from Universidade Federal do Rio de Janeiro.

### PBMC cultures and cytokine measurements

Peripheral blood mononuclear cells (PBMCs) were isolated from heparinized blood by Ficoll gradient. Briefly, blood layered on Ficoll-Paque plus (GE-Healthcare) were centrifuged 410 x *g*, 40 minutes, RT. One-million cells per well were distributed in 96-well flat-bottom plates and incubated in complete media (RPMI 1640, 10% FBS, 100 mg/ml streptomycin, 100 U/ml penicillin) with 5 μg/mL from either N, RBD and S recombinant antigen or anti-CD3 (1 μg/mL) and anti-CD28 (0.5 μg/mL) as positive controls. Unstimulated cells were used to assess the background production of cytokines. Culture supernatants were harvested after 72 h and frozen at −80 °C until analysis. Levels of IFN-γ were measured by ELISA following the manufacturer's protocol (BD, OptEIA Human IFN-γ, Cat 555142). Alternatively, cytokines in the supernatant were measured by Cytometric Bead Array (CBA), through the kits Human Inflammatory CBA and Human Th1/Th2/Th17 CBA (BD Biosciences, Cat 551811 and 560484, respectively), following the manufacturer's instructions. The samples were read and analyzed on FACSVerse (BD Biosciences).

### Flow cytometry of human samples

PBMCs from healthy donors, convalescents or Coronavac-vaccinated donors were thawed in RPMI 1640 (Sigma-Aldrich) benzonase nuclease (20 U/mL, Sigma). Cells were plated in complete media (RPMI 1640, 10% FBS, 100 mg/ml streptomycin, 100 U/ml penicillin), rested for 5 h and then incubated with positive controls (anti-CD3, 1 μg/mL and anti-CD28, 0.5 μg/mL, BD), negative controls (SpiN Vehicle) or 20 μg/mL of SpiN in 5% $CO_2$ at 37 °C. After 12 h incubation, 2.5 μg/mL each of BFA and Monensin was added for additional 6 h. After total 18 h incubation cells were harvested and stained for viability (7-AAD, BD Pharmingen) and surface antigens anti-CD4 (BV605, RPA-T4, BD), anti-CD8 (AlexaFluor700, SK1, Biolegend), anti-CD45RO (BV786, UCHL1, BD) and anti-CD27 (APC-Cy7, O323, Biolegend). After washing, cells were fixed and permeabilized according to the manufacturer's instructions (FoxP3 staining buffer set, eBioscience). Cells were then stained for the intracellular antigens anti-IFN-γ (PE-Cy7, 4 S.B3, eBioscience), anti-CD3 (FITC, UCHT1, BD) anti-CD69 (BV421, FN50, Biolegend) and anti-TNF (APC, Mab11, eBioscience). Samples were acquired on an LSR-FORTESSA and analyzed on FlowJo. T-cell subpopulations were gated on viable CD3$^+$CD4$^+$ and CD3$^+$CD8$^+$ events. More detailed analysis of naive and memory sub-populations were done based on CD27/CD69/CD45RO expression and intracellular IFN-γ: effector memory (EM, CD45RO$^+$CD27$^-$), central memory (CM, CD45RO$^+$CD27$^+$), effector (Eff, CD45RO$^-$CD27$^-$), and naïve (Nv, CD45RO$^-$CD27$^+$) cells. Dimensionality reduction analysis was done using default settings of Uniform Manifold Approximation and Projection (UMAP) plugin implemented in FlowJo.

### Detection of human, mouse and hamster antigen-specific antibodies

Plates were coated overnight with 0.4 μg/well of either N, RBD, S recombinant proteins, or alternatively 10$^4$ PFU/well of UV-inactivated SARS-CoV-2, and blocked for 2 h with PBS containing 2% bovine serum albumin (PBS-2% BSA) at 37 °C. Serum was serially diluted, and the bronchoalveolar lavage (BALF) samples were tested at 1:1 dilution in PBS-2% BSA and incubated for 1 h at 37 °C, and then incubated with anti-human IgG-HRP antibody (Fapon), anti-hamster IgG-HRP or anti-mouse total IgG, IgG1, IgG2c conjugated with streptavidin-HRP (Southern Biotech), all diluted 1:5000. After 5 washes, plates were revealed with 1-Step ultra TMB substrate solution (Biolegend) for 15 minutes in the dark, and reaction was stopped by adding 2 N $H_2SO_4$ (Sigma). Plates were read in 450 nm and results were expressed as raw optical density (OD). The antibody titer was determined by the sera dilution that yielded 50% of the maximum antibody reactivity to the antigen in the ELISA.

## Reduction neutralization assay

One day prior to infection, $10^5$ Vero E6 cells were seeded in Dulbecco's modified eagle media (DMEM) (Vitrocell, Brazil) with 10% Fetal Bovine Serum (FBS) to each well of a 48 wells plate. On the next day, sera samples from mice or human were heat inactivated by incubation at 56 °C for 1 h on a warm bath. Samples were two-fold serially diluted (1:10 to 1:320 (v/v)) in DMEM and mixed with 100 PFUs of SARS-CoV-2 viral stock. Media only was used instead of sera samples for positive control. The mixture was incubated for 1 h ta 37 °C to allow antibody binding to the viral particles. Next, Vero E6 cell culture supernatant was removed and the cells were inoculated with 50 µl/well of the sera-virus mixture, incubated for 1 h at room temperature under gently rocking to allow viral-biding to cells. Then, 1 ml of pre-warmed DMEM with 2% FBS and 2% of carboxymethylcellulose (CMC) was gently added to each well and the plates were incubated at 37 °C and 5% $CO_2$ atmosphere for 4 days to allow viral plaque formation. Then, the cells were fixed with 4% formaldehyde solution diluted in PBS for 2 h and stained with 1% Naphtol blue black (Sigma, USA) solution for 1 h for plaque visualization. Neutralization activity was determined by plaque numbers reduction compared to the positive control.

## Viral quantitative reverse transcription polymerase chain reaction

Total RNA was extracted from homogenized mice tissues using the RNeasy Mini Kit (Qiagen, Cat 74104), according to protocols provided by the manufacturers. qRT-PCR was performed in 12 µL reactions using GoTaq Probe 1-step RT-qPCR System (Promega, US) according to the manufacturer's instructions, using 75 ng of total RNA per reaction. Primers and fluorescent probes were designed based on previously described diagnostic qRT-PCR protocol specific for SARS-CoV-2, which amplify a 100 bp amplicon from the E gene of SARS-CoV-2[72]. Probe FAM-ACACTAGCCATCCT-TACTGCGCTTCG-BBQ, F 5' ACAGGTACGTTAATAGTTAATAGCGT 3', R 5' ATATTGCAGCAGTACGCACACA 3'. Cycling conditions were 45 °C for 15 min and 95 °C for 3 min followed by 45 cycles at 95 °C for 15 s and 58 °C for 60 s, using Quantstudio 5 Real-Time PCR system (Applied Biosystems, USA). For viral load quantification, a standard curve based on a plasmid containing the E gene sequence (SARS-CoV-2 Wuhan-Hu isolate sequence) was constructed. Serial 10-fold dilutions of plasmid DNA that correspond to viral copies ranging from 2 to $2 \times 10^5$ were used as templates to prepare the standard curves. Real-time PCR assays were carried out in triplicate and the resulting Ct values by plotted against the copy number of the viral genome.

## Immunization, challenge and histopathology

Hamsters and C57BL/6 or hACE2 mice received two administrations at 21 days apart, containing 10 µg of RBD, N or SpiN adjuvanted with 50 µg of Hiltonol (Poly ICLC, supplied by Oncovir, Washington, D.C.)[40,42]. Alternatively, they received one dose of a non-replicating chimpanzee adenovirus encoding the S protein from the Wuhan SARS-CoV-2 (Covishield) at a concentration of $10^{10}$ PFU/mice. The solutions were inoculated intramuscularly in a final volume of 50 µL into each tibial muscle. Thirty days post immunization, animals were challenged intranasally with $5 \times 10^4$ PFU of SARS-CoV-2 Wuhan, Delta or $2.5 \times 10^4$ of Omicron isolates (for hACE2 mice) and $10^5$ PFU for hamsters. The body weight, clinical signs and survival were evaluated for 11 days post-infection. For the histopathology analyses, harvested tissues were fixed in phosphate-buffered 10% formalin for seven days, embedded in paraffin, processed using a Tissue processor PT05 TS (LUPETEC, UK) tissue processor and embedded in histological paraffin (Histosec, Sigma-Aldrich). The 4 µm thick sections were stained with hematoxylin and eosin.

## $CD8^+$ and $CD4^+$ T cells depletion and antibody passive transfer

K18-hACE2 mice immunized with either SpiN + Poly ICLC or Covishield were treated via i.p. with either 0.5 mg/mouse of rat anti-mouse CD8a or CD4 mAbs (BioXCell, clone 2.43, and GK1.5, respectively), or both. Control groups received 0.2 mg/mouse of isotype control rat anti-KLH IgG (clone LTF-2, BioXCell). The treatment was carried out on days −3, −2, and −1 before the challenge. Depletion was confirmed by flow cytometry analysis of whole blood before infection. For antibody passive transfer, non-immunized K18-hACE2 mice were administered with 200 µL of sera from mice administered with either SpiN + Poly ICLC, Covishield or with Poly ICLC only. The inoculation was performed via i.p. one-day prior challenge. Antibodies' titer anti-N and RBD was confirmed by ELISA.

## Measurements of mouse cytokines

Mouse splenocytes were isolated by macerating the spleen through a 100 µm pore cell strainer (Cell Strainer, BD Falcon) followed by treatment with ACK buffer for erythrocytes lysis. The number of cells was adjusted to $10^6$ cells per well and then stimulated with 10 µg/mL of RBD or N. Concanavalin A (Sigma, 5 µg/mL) was used as the positive control. The supernatants were collected 72 h post-stimulation and the levels of IFN-γ and IL-10 were determined by ELISA (R&D Systems, Cat DY485 and DY417, respectively). Alternatively, cytokines were measured in culture supernatant through the Mouse Th1/Th2/Th17 CBA (BD Biosciences, Cat 560485), following the manufacturer's instructions. The samples were read and analyzed on FACSVerse (BD Biosciences).

## Cytokines and chemokines measurements by qRT-PCR

The RNA samples isolated from the lungs of immunized and challenged mice at 5 dpi were treated with DNase (Promega), and then converted into cDNA using the High-Capacity cDNA Reverse Transcription Kit (Applied Biosystems, Cat 4368814), according to the manufacturer's instructions. qPCRs reactions were performed with Sybr Green PCR Master Mix (Applied Biosystems) in an ABI7500 Real-Time PCR System (Applied Biosystems) under standard conditions. Primer sequences are presented in Supplementary Table 6. qRT-PCR data were presented as 1/ΔCT.

## Immunofluorescence assays

A 16-well chamber slide (ThermoFisher) was coated with $10^4$ Vero E6 cells/well and incubated overnight with SARS-CoV-2 in a multiplicity of infection (M.O.I) of 10. Then, the cells were treated with Brefeldin A for 4 h, fixed with paraformaldehyde 4% and permeabilized with PBS-P (PBS 0.5% BSA + 0.5% saponin). Later, the wells were blocked with BSA 1% and incubated with sera from mice immunized with RBD, N or SpiN. The secondary antibody Alexa Fluor 594 anti-mouse IgG (Thermo-Fisher) was added and the nucleus was stained with DAPI (Thermo-Fisher). The slides were analyzed in the confocal microscope LSM 780 Carl Zeiss AxioObserver, objective x63 oil NA 1.4. Images were processed with the software ImageJ version 2.1 for Mac.

## Flow cytometry of mouse splenocytes and lungs

For imunophenotyping, a total of $2 \times 10^6$ of splenocytes derived from immunized mice were incubated for 18 h at 37 °C and 5% $CO_2$ with RPMI 1640 medium alone or containing 10 µg/mL of SpiN, RBD or N proteins. During the last 6 h of culture, GolgiStop and GolgiPlug Protein Transport Inhibitors (BD Biosciences) were added to the cell cultures. The splenocytes were then washed with PBS, stained with Live/Dead reagent (Invitrogen) and incubated with FcBlock (BD Biosciences)[73]. The following mAbs were used to label cell surface markers: anti-CD3 PerCP-Cy5.5 (BD, clone 145-2C11) or FITC (BD, clone 145-2C11), anti-CD4 Alexa Fluor 700 (Invitrogen, clone KG1.5), anti-CD8 APC-Cy7 (Biolegend, clone 53-6.7), anti-CD62L PE-Cy7 and anti-CD44 APC (Invitrogen, clone IM7). For intracellular staining, cells were

washed, fixed and permeabilized according to the manufacturer's instructions (Cytofix/Cytoperm, BD Biosciences) and stained with anti-IFNγ PERCP (eBioscience, XMG1.2). Flow cytometry was carried out using a BD LSRFortessa with BD FACSDIVA V8.0.1 software and ~100,000 live CD3⁺ cells were acquired. Data were analyzed using FlowJo v10.5.3 software.

For flow cytometry staining, lungs were excised, minced with scissors, and enzyme-digested using 2 mg/ml of collagenase IV (Sigma) diluted in 1 mL of RPMI. The suspensions were incubated at 37 °C for 30 min with regular shake. Tissue fragments were filtered using 50 μm pore size nylon filter Filcon system (BD Biosciences) and then centrifuged. The supernatants were discarded and erythrocytes in the cell pellets were lysed using ACK solution. The remaining cells were resuspended in RPMI 5% FBS, counted in Neubauer chamber, washed with PBS 1x and used for extracellular flow cytometry staining. Cells were then fixed, permeabilized and stained with the following antibodies: Zombie Aqua Fixable Viability Kit (Biolegend, Cat 423101), anti-CD11b APC-Cy7 (M1/70, Biolegend), anti-CD11c PE-Cy7 (N418, Biolegend), anti-Ly6C PERCP (HK1.4, Biolegend), anti-MHC-II FITC (M5/11.15.2, Biolegend) and anti-Ly6G APC (1A8, Biolegend). Alternatively, cells were cultured with SpiN protein and stained with anti-CD3 FITC (145-2C11, Biolegend) anti-CD4 APC-Cy7 (GK1.5, BD), anti-CD8 PERCP (53–6.7, Biolegend), anti-CD69 PE (H1.2F3, Biolegend), anti-CD103 BV421 (2E7, Biolegend), anti-IFN-γ APC (XMG1.2, Biolegend) and anti-TNF PE-Cy7 (MP6-XT22, Biolegend). Flow cytometry was carried out using a BD LSRFortessa, and ~100,000 live cells were acquired. Data were analyzed using FlowJo v10.5.3 software.

## Statistical analysis

Statistical analysis was conducted using GraphPad Prism 6.0 for Mac (GraphPad Inc, USA). First, outliers were detected with Grubbs's test and then D'Agostino-Pearson was run to verify data normality. The tests used on each data analysis are explained on figure legends. In general, comparison between the groups was performed through unpaired *t*-test or Mann–Whitney U test, according to data distribution. Weight measurements and flow cytometry data were analyzed by Two-Way ANOVA, followed by Sidak's multiple comparisons test. For survival analysis, the log-rank test was used. Statistical differences were considered significant when p values ≤0.05.

## Reporting summary

Further information on research design is available in the Nature Research Reporting Summary linked to this article.

## Data availability

The authors declare that all data supporting the findings of this study are available within the paper and its supplementary information files. For epitope prediction of Spike (6VSB https://doi.org/10.2210/pdb6VSB/pdb) and Nucleocapsid (7SD4_1 10.2210/pdb7SD4/pdb) proteins, we used the immune epitope database (IEDB). For the needle plot, we used Spike and Nucleocapsid amino acid sequences recovered from the alignment of the reference genomes of Alpha (MZ344997), Beta (MW598419.1), Delta (MZ359831.1), Gamma (MZ169911.1), Omicron BA.1 (OL672836.1), Omicron BA.2 (PRJNA784038), and the original lineage (B1 – Wuhan – EPI_ISL_402123 https://www.epicov.org/epi3/frontend#4c5e54). Source data for Figs. 1f, g; 2b–p; 3a–f; 4a–g; 5a–c; 6a–e, g–o; 7a–h; 8a–d are provided with the paper. If any more information is needed, data are available from the corresponding author upon reasonable request. Source data are provided with this paper.

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

## Acknowledgements

This work was funded by Rede Virus from the Ministry of Science, Technology and Innovation, (Finep 01.20.0010 and 01.20.0005.00; CNPq 403514/2020-7 and 403701/2020-1 R.T.G.); National Institute of

Science and Technology of Vaccines (Fapemig APQ-03608-17/CNPq 465293/2014-0 R.T.G.); FAPESP (2020/05527-0 R.T.G.); Ministry of Education (CAPES); Fundação Oswaldo Cruz (INOVA, 2020), Prefeitura de Belo Horizonte; as well as Parliamentary Amendment of State and Federal Representatives from Minas Gerais. We thank our project analyst Ms. Cristiane Gomes, our secretary Elizabeth Araújo, and the technicians Ms. Franciele Pioto, Rosângela Pereira and Rodrigo Moreli.

## Author contributions

J.C. performed epitope predictions. R.M. was responsible for the construction of the needle plot. J.C., G.B. and D.D. contributed to protein's conception and plasmids constructions. N.S. and B.C. performed protein expression and purification, SDS-PAGE and Western Blots. G.G.A., L.I.O., T.G.M., and M.A. performed IgG and IFN-γ ELISAS of human samples. G.A., B.V., J.C., and L.A. performed CBA and flow cytometry of human samples. J.C., N.S.H-S., and P.A. were responsible for mice and hamsters immunizations and ELISAS. J.C. performed flow cytometry of mice spleens and BALF. J.C., M.J.F. P.A., and O.C.N. performed the challenge experiments with SARS-CoV-2. M.J.F. performed the neutralization assays. J.C. and M.J.F. performed immunofluorescence assays. J.C., L.F., and A.F. performed viral RNA extraction and quantification by RT-PCR. J.C. performed cytokine and chemokine mRNA measurements by RT-PCR. M.P. and L.B. were responsible for flow cytometry of mice lungs. B.R. and S.G.R. performed histopathological analysis. E.D. provided SARS-CoV-2 viruses. A.S. and O.C. provided reagents. J.C., M.J.F., R.T.G, S.R.T., A.P.F., F.F., H.S. J.S.S., A.M., and J.K. contributed to study's conception and design. J.C. and R.T.G. wrote the manuscript. All authors discussed the results and the manuscript.

## Competing interests

R.T.G., S.R.T., A.P.F., F.F., N.S., J.C., N.S.H-S., and P.A. are co-inventors of the potential COVID-19 vaccine evaluated in this study. The patent is under evaluation process, application number BR1020210095733. The other authors declare no competing interests.

## Additional information

[1]Centro de Tecnologia de Vacinas, Universidade Federal de Minas Gerais, Parque Tecnológico de Belo Horizonte 31.310-260 MG, Brazil. [2]Fundação Oswaldo Cruz-Minas, Belo Horizonte 30.190-002 MG, Brazil. [3]Plataforma de Medicina Translacional da Fundação Oswaldo Cruz e Faculdade de Medicina de Ribeirão Preto, Universidade de São Paulo, Ribeirão Preto 14.040-030 SP, Brazil. [4]Fundação Hospitalar do Estado de Minas Gerais, Belo Horizonte 31.630-901 MG, Brazil. [5]Instituto do Coração, Universidade de São Paulo, São Paulo 05403-900 SP, Brazil. [6]Instituto de Ciências Biológicas, Universidade de São Paulo, São Paulo 05508-000 SP, Brazil. [7]Oncovir, Inc; Orygen, Biotecnologia, São Paulo 04538-133 SP, Brazil. [8]Instituto de Ciências Biológicas, Universidade Federal de Minas Gerais, Belo Horizonte 31.270-901 MG, Brazil. [9]Faculdade de Farmácia, Universidade Federal de Minas Gerais, Belo Horizonte 31.270-901 MG, Brazil. [10]University of Massachusetts Medical School, Worcester 01605 MA, USA. ✉e-mail: ricardo.gazzinelli@umassmed.edu

