## [Peer Review File · Nature Communications]

Immunization with a RBD/Nucleocapsid fusion protein promotes neutralizing antibody-independent resistance to infection with the wild type and SARS-CoV-2 variants of concernThis manuscript has been previously reviewed at another journal that is not operating a transparent peer review scheme. This document only contains reviewer comments and rebuttal letters for versions considered at *Nature Communications*.

REVIEWERS' COMMENTS

Reviewer #1 (Remarks to the Author):

The authors addressed the raised question and concerns by new data and by optimising the data presentation.

Concerning the last point on definition of the specific T cell epitopes I would suggest to include in the discussion section that this has to be further investigated.

Reviewer #2 (Remarks to the Author):

The authors have satisfactorily addressed my comments, and the paper is now greatly improved, and should be published in NATURE COMMUNICATIONS.

REVIEWERS' COMMENTS

Reviewer #1 (Remarks to the Author):

The authors addressed the raised question and concerns by new data and by optimising the data presentation.

Concerning the last point on definition of the specific T cell epitopes I would suggest to include in the discussion section that this has to be further investigated.

It was included in the last paragraph of the discussion.

Reviewer #2 (Remarks to the Author):

The authors have satisfactorily addressed my comments, and the paper is now greatly improved, and should be published in NATURE COMMUNICATIONS.